# Evidence-based vs. social media based high-intensity interval training protocols: Physiological and perceptual responses

**Katie L. Hesketh**[1], **Hannah Church**[1], **Florence Kinnafick**[2], **Sam O. Shepherd**[1], **Anton J. M. Wagenmakers**[1], **Matthew Cocks**[1]*, **Juliette A. Strauss**[1]

**1** Research Institute for Sport and Exercise Science, Liverpool John Moores University, Liverpool, United Kingdom, **2** School of Sport, Exercise and Health Sciences, National Centre for Sport and Exercise Medicine, Loughborough University, Loughborough, United Kingdom

* M.S.Cocks@ljmu.ac.uk

**Data Availability Statement:** All relevant data are within the paper, and files are available in the LJMU

## Abstract

### Objective

High intensity interval training (HIIT) is a time-efficient exercise modality to improve cardio-respiratory fitness, and has recently been popularised by social media influencers. However, little is known regarding acute physiological and perceptual responses to these online protocols compared to HIIT protocols used within research. The aim was to investigate acute physiological, perceptual and motivational responses to two HIIT protocols popular on social media, and compare these to two evidence-based protocols.

### Methods

Twenty-seven recreationally active (>1 exercise session /week) participants (Age: 22±3y, BMI: 24.3±2.4) completed a randomised cross-over study, whereby each participant completed four HIIT protocols, two already established in research (Ergo-60:60 (cycling 10x60s at 100%$W_{max}$ with 60s rest), BW-60:60 (body-weight exercises 10x60s with 60s rest)) and two promoted on social media (SM-20:10 (body-weight exercises 20x20s with 10s rest) and SM-40:20 (body-weight exercises 15x40s with 20s rest)). Blood lactate, heart rate (HR), feeling scale (FS), felt arousal scale (FSA), enjoyment and perceived competence were measured in response to each protocol.

### Results

Significant differences were observed between BW-60:60 and SM-20:10 for the proportion of intervals meeting the ACSM high-intensity exercise criterion (>80% of $HR_{max}$) (BW-60:60 93±10%, SM-20:10 74±20%, $P = 0.039$) and change in lactate (BW-60:60 +7.8±3.7mmol/L, SM-20:10 +5.5±2.6mmol/L, $P = 0.001$). The percentage of time spent above the criterion HR was also significantly lower in SM-20:10 compared to all other protocols (Ergo-60:60 13.9±4.9min, BW-60:60 13.5±3.5min, SM-40:20 12.1±2.4min, SM-20:10 7.7±3.1, $P<0.05$). No differences were observed in lowest reported FS between protocols ($P = 0.268$), but FS decreased linearly throughout Ergo-60:60 and BW-60:60 (first vs. last interval P<0.05), but

database (DOI: https://doi.org/10.24377/LJMU.d.00000099).

**Funding:** The authors received no specific funding for this work.

**Competing interests:** The authors have declared that no competing interests exist.

not in SM-20:10 or SM-40:20 ($P>0.05$). Enjoyment was higher upon completion of BW-60:60 compared to Ergo-60:60 and SM-40:20 ($P<0.05$).

## Conclusions

This study shows that HIIT protocols available on social media offer an interesting real-world alternative for promoting exercise participation. Future studies should continue to investigate these highly popular and practical HIIT protocols.

## Introduction

High intensity interval training includes brief, intermittent bursts of vigorous activity (typically between 80–100% $HR_{max}$), interspersed by periods of rest or recovery [1]. It is well established that HIIT is an effective time efficient means of training, resulting in equal or superior physiological adaptations to traditional moderate-intensity continuous training (MICT), despite substantially lower training volumes [2]. Following the positive reporting of this research through established media outlets, HIIT topped the American College of Sports Medicine's (ACSM) Worldwide Fitness Trends list for the first time in 2014, and has remained in the top three since; returning to first place in 2018 [3–5]. In addition to the promotion of HIIT through the established media, its popularity has grown through endorsements by social media 'influencers' and the availability of fitness videos on media sharing sites. For example, one of the most popular HIIT exercise videos available on YouTube has over 15 million views. These social media influencers provide interesting opportunities to engage with audiences on a personal level, and can assist in the delivery of health improvement interventions [6]. Even though social media outlets have helped to establish HIIT as a popular training mode, there is no research comparing the protocols used in social media videos to those employed within the research. Although, one study has compared the acute responses to a video on a popular smartphone application (The 7-minute Workout (12x30s with 10s recovery)) with the same protocol carried out on a cycle ergometer, reporting greater mean and peak $VO_2$, heart rate (HR) and rate of perceived exertion following the cycling modality [7]. Importantly, the protocols promoted by social media influencers often employ interval durations and/or work-to-rest ratios that have not been backed by published research. Furthermore, within the peer-reviewed scientific research HIIT has primarily been developed as a time-efficient protocol to increase cardiorespiratory fitness. As such the exercise modalities used have been mainly aerobic in nature (e.g. running, cycling or body weight exercises using jumps) with the aim of eliciting a HR $\geq 80\% HR_{max.}$ In contrast, protocols used within social media HIIT often include resistance-based exercises (e.g. press ups).

Recent work has suggested that the acute physiological response to HIIT may influence its long-term effectiveness. Importantly, Fiorenza et al. [8] demonstrated that metabolic stress is a key mediator of the acute molecular response to HIIT in endurance trained cyclists [8]. In mice, Hoshino et al. [9] suggested that repeated lactate accumulation during HIIT may be associated with training-induced mitochondrial adaptation. Furthermore Moholdt et al. [10], demonstrated that the mean HR achieved during HIIT intervals is central to long term increases in $VO_{2peak}$ achieved following training, in patients with coronary heart disease. Taken together this data suggests that adaptation to long-term exercise training could be dependent on the magnitude of the stimulus received during each acute bout of exercise.

Although acute physiological responses are an important determinant of long-term adaptation, perceptual responses (positive/negative affect) during exercise and factors related to

motivation (enjoyment and perceived competence) during and following exercise also influence the long term effectiveness of a training programme, as these factors can predict exercise adherence [10]. As such, assessing the acute psychological responses (i.e. how one is feeling during a HIIT session (affect)) to different high intensity interval exercise (HIIT) protocols may provide important information regarding future effectiveness. It has been hypothesised that the strenuous nature of HIIT may be a barrier to participation, as individuals are likely to avoid exercise if it is found to be aversive [11]. This assumption is based upon Dual Mode Theory proposed by Ekkekakis [12], which argues pleasure (affect) experienced during exercise declines when individuals exercise above ventilatory threshold. Therefore, assessing the affective response (feelings of pleasure/ displeasure) to HIIT protocols is important, as negative affect during exercise can act as a deterrent [13], while pleasurable experience is a determinant of exercise participation [14]. However, the majority of work used to support Dual Mode Theory has been carried out using continuous high intensity exercise, and its use within HIIT has been critiqued previously [15]. Motivation is well known to be a determinant of physical activity participation, a macro-theory of motivation which has been used to explain physical activity behaviour is Self Determination Theory [16]. Self Determination Theory proposes that motivation arises from the satisfaction of basic psychological needs (autonomy, competence and relatedness) [17]. It is the satisfaction of these basic psychological needs that have been shown to predict regular exercise participation [18]. As such, if individuals do not possess perceptions of competence during a HIIT protocol, they are more likely to disengage and not adhere to a programme. Finally, Stork and Martin Ginis [19] hypothesised that enjoyment predicts attitudes towards HIIT, which in turn mediate future intentions to participate.

Therefore, the aim of this study was to investigate the acute physiological, perceptual and motivational responses to two HIIT protocols popular on social media (SM-20:10, SM-40:20), and compare these to two evidence based HIIT protocols.

## Methods

### Participants

Twenty-seven recreationally active (defined as completing >1 but <4 structured exercise sessions per week) participants (male/female: n = 13/14, age: 22±3y, height: 1.70±0.09m, weight: 70.4±11.2kg, BMI: 24.3±2.4, $VO_{2peak}$: 42.2±7.2 ml.min$^{-1}$.kg$^{-1}$) were recruited from Liverpool John Moores University via internal email and posters. Exclusion criteria were those with a known cardiovascular or metabolic disease, pregnant or breastfeeding women, and those currently carrying an injury. The study was approved by the Liverpool John Moores Research Ethics Committee, and all participants gave written informed consent to the protocol which conformed to the Declaration of Helsinki.

### Study design

The study used a randomized, counter-balanced crossover design to investigate the four HIIT protocols. Participants attended an initial experimental visit followed by 4 experimental trials to assess the acute physiological and psychological responses to the HIIT protocols. All visits were performed within the same laboratory environment at a similar time of day (between 11am and 3pm). All participants were asked to maintain their regular diet, to refrain from vigorous exercise 24 hours before each session and not to eat 3 hours before. All visits were separated by at least 48 hours. Participants were not familiarised to the HIIT protocols before performing them in the experimental trials.

### Initial experimental visit

Prior to the experimental trials participants completed an incremental exercise test to exhaustion on an electronically braked cycle ergometer (Lode Corival, The Netherlands), to determine $VO_{2peak}$, maximum heart rate and maximal aerobic power output ($W_{max}$). The method is described fully by Scott et al. [16], but briefly, participants began cycling at 25 W for females and 60 W for males for 3 min; following this the workload was increased by 35 W every 3 min until volitional fatigue. $VO_{2peak}$ was assessed using an online gas collection system (Metamax 3B, Cortex, Germany) and was defined as the highest value achieved over a 15 second recording period. HR was monitored throughout the test (Polar H10, Kempele, Finland).

### Experimental visits

All experimental visits were identical except for the HIIT protocol performed. Prior to exercise a capillary blood sample was obtained from a fingertip for an immediate assessment of blood lactate (Biosen, EKD diagnostics, UK). Participants were introduced to the Feeling Scale and Felt Arousal Scale [20]. Scores on each scale were recorded immediately before and after each interval to indicate responses during the interval and at rest. Before starting the protocols all participants completed a 2-minute warm up; either 25W on a cycle ergometer (Ergo-60:60) or jogging on the spot (BW-60:60, SM-20:10 and SM-40:20). Participants were given no encouragement by the research team during the protocols, but if an exercise was being conducted incorrectly the researcher would advise/demonstrate to ensure consistency and minimise injury risk. HR was measured continuously throughout the exercise protocols (Polar H10). Following completion of the protocols (within ~1min) a post exercise blood lactate was collected. Finally, 10 minutes after completion of the protocol all participants were asked to complete the Intrinsic Motivation Inventory (IMI) [21].

### Training protocols

Acute measurements were collected across four different HIIT protocols; two were evidence-based and two were accessed via the social media outlet YouTube. The Ergo-60:60 protocol has been successfully used to increase cardiorespiratory fitness in a variety of populations over a 2–12 week period (Sedentary [22], obese individuals [23], individuals with type 2 diabetes [24]). Recently this protocol has be adapted for the home environment using body-weight exercises. This adapted protocol has also been shown to induce increases in cardiorespiratory fitness in a variety of populations (sedentary [25], people with elevated cardiovascular disease risk [26], people with type 1 diabetes [27]). There are countless videos featuring HIIT on social media, as such, our aim was to choose two protocols which we felt were representative of the field. To assess the most common protocols used on social media the protocols had to meet the following criteria 1) be featured on a popular YouTube fitness channel 2) have 'HIIT' in the title of the video 3) take less than 20 minutes, to take advantage of the time-saving nature of HIIT 4) include body weight exercises with no equipment. The SM-20:10 protocol uses "Tabata training", a variation of the original protocol designed by Tabata et al. [28] which has been demonstrated to increase $VO_{2peak}$. This was included as variations of "Tabata Training" are popular within social media. SM-40:20 was included as the protocol used a blend of aerobic and resistance-based exercises (e.g. press-ups) which would not typically fall under the traditional definition of HIIT, but is used by a number of videos found on social media channels.

**Ergometer laboratory based HIIT (Ergo-60:60).** The laboratory-based HIIT protocol was completed on a cycle ergometer (Lode Corival), and consisted of repeated 60 second efforts of high intensity cycling at 100% $W_{max}$ (obtained from the incremental exercise test) [29]. These intervals were interspersed by 60 seconds of cycling at a low intensity (50 W).

**Table 1. Summary of protocols used to measure acute responses to HIIT.**

| ' | Number of intervals | Intensity of intervals | Interval duration (seconds) | Rest duration (seconds) | Total duration (minutes) | Work: Rest Ratio | Exercise |
|---|---|---|---|---|---|---|---|
| Ergo-60:60 | 10 | 100% Wmax | 60 | 60 | 20 | 1:1 | Cycling |
| BW-60:60 | 10 | As many repetitions as possible | 60 | 60 | 20 | 1:1 | **1)** mountain climbers + lateral jumps **2)** floor jacks + get ups **3)** squat thrusts + elbow to knee **4)** split squats + jogging boxers **5)** burpees + jogging on the spot **6)** jogging with high knees + squat jumps **7)** spotty dogs + X jumps **8)** jump overs + jumping jacks **9)** tuck jumps + clapping jacks **10)** mountain climbers + lateral jumps |
| SM-20:10 | 20 | Guided by exercise video | 20 | 10 (20s between sets) | 11.5 | 2:1 | **1)** Broad jumps x2 jumping jacks **2)** pop squats **3)** burpees with a kick **4)** 3 jumps and lunge **5)** squat jump slides |
| SM-40:20 | 15 | Guided by exercise video | 40 | 20 | 15 | 2:1 | **1)** walkout press-up with shoulder taps **2)** squat with knee to elbow *left* **3)** 8 high knees and burpee **4)** squat with knee to elbow *right* **5)** kick through **6)** knee to elbow plank **7)** 90˚ squat jump **8)** staggered stance push up *right* **9)** jogging with punches **10)** staggered stance push up *left* **11)** side lunge *right* **12)** bear crawl **13)** side lunge *left* **14)** narrow push up with arm lift **15)** 180∘ burpee |

Ergo-60:60 10x60s on a cycle ergometer, with 60s rest. BW-60:60 10x 60s body weight exercises, with 60s rest. SM-20:10 20x 10s with 20s rest, exercises provided from a social media video. SM-40:20 15x 40s with 10s rest, exercises provided from a social media video.

Subjects completed ten high-intensity intervals. The total time commitment for the protocol (excluding warm-up) was 20 minutes.

**Home-based body weight HIIT (BW-60:60).** The established body weight exercise protocol was identical to that used in Ergo-60:60, 10 repeated 60 second bouts of high intensity exercise, interspersed with 60 seconds of rest [26]. The 60 second intervals were comprised of two different bodyweight exercises performed for 30 seconds each, with no rest in between. Prior to the protocol participants were given 10 exercise pairs, which were verbally explained and demonstrated by the research team. All participants completed the same exercise pairs, 9 pairs were used with one pair completed twice (see **Table 1**). Participants were asked to complete as many repetitions as possible in 60 seconds. The total time commitment for the protocol (excluding warm-up) was 20 minutes.

**Social media HIIT 1 (SM-20:10).** Participants followed the video https://www.youtube.com/watch?v=VhdXXqcoco0 available via the Fitness Blender YouTube Channel. The video was shown (with sound) on a television screen. The protocol consisted of 5 sets of exercise. Each set used a different exercise and was made up of 4x20s intervals, separated by 10 seconds of rest (see **Table 1**). Each set was then separated by 20s of rest. The total time commitment for the protocol (excluding warm-up) was 11.5 minutes.

**Social media HIIT 2 (SM-40:20).** Participants followed the YouTube video https://www.youtube.com/watch?v=yz59KggOtb0) available via the Body Coach TV YouTube Channel. The video was shown on a television screen with the volume on. The protocol involved 15x40s intervals, separated by 20 seconds rest, a different exercise was used for every interval (15 exercises in total, see **Table 1**). The total time commitment for the protocol (excluding warm-up) was 15 minutes.

### Assessment of heart rate during exercise

HR was assessed continuously throughout each protocol (Polar H10). The time of the start and end of each interval were written down and used to denote the start and end of each interval during analysis. Following each exercise session, HR data was immediately downloaded to excel for offline analysis and has been presented as a % of HRmax achieved on the incremental

exercise test. Mean HR for the whole session (session $HR_{mean}$), and the highest HR achieved during each session were determined (session $HR_{peak}$). Mean and peak HR ($HR_{mean}$ and $HR_{peak}$) were also determined for every interval. Mean values for each exercise session were then calculated and used to determine the interval $HR_{peak}$ and interval $HR_{mean}$. The ACSM suggests that HIIT should be performed at a HR above 80% of an individual's $HR_{max}$ [30]. As such, we determined the proportion of intervals meeting the high-intensity criterion (HR >80% of max) and time spent above the criterion HR, as suggested by Taylor et al. [31].

## Perceptual responses during exercise

**Feeling scale and felt arousal scale.** The Feeling Scale is an 11-point scale ranging from +5 to -5 [32] and is commonly used to measure affect responses (pleasure/displeasure) during exercise [14,33]. The scale presents the following verbal anchors: -5 = very bad; -3 = bad; -1 = fairly bad; 0 = neutral; +1 fairly good; +3 = good; and +5 = very good. The Felt Arousal Scale measures perceived activation along a 6-point scale ranging from low arousal (1) to high arousal (6). All participants were given standardised instructions on how to use the scale and verbal anchors were provided by one member of the research team. The participants were asked their score on each of the scales, based on their feelings at the time of completion, immediately before and after each interval.

## Motivation

**Intrinsic motivation inventory.** The Intrinsic Motivation Inventory (IMI) is a multidimensional measurement device, which includes two subscales to assess self-reported interest/enjoyment and perceived competence. The IMI had a reported Cronbach's alpha coefficient of 0.92 for both the interest/enjoyment scales and the perceived competence scales. All participants were asked to read the phrases in the two subscales (13 in total), and were asked to rate them on a Likert scale from 0 (not true at all) to 7 (very true). The two subscale scores were then calculated by averaging across all the items on the subscale.

## Data analysis

Data is expressed as means ± SD and was analysed using SPSS Version 26.0 (Chicago, IL, USA). One subject was not able to finish the Ergo-60:60 protocol due to fatigue, therefore, the data from this participant was removed during analysis (n = 26). A one-way within subject ANOVA was used to investigate differences between protocols, for heart rate responses during exercise, change in lactate, lowest recorded score on the Feeling Scale and responses to the IMI (interest/enjoyment and perceived competence). A one-way between-subjects ANOVA was also used to assess responses to the Feeling Scale over time within each HIIT protocol. Partial eta squared (η2) was used as an estimate of effect size, with a small effect = 0.01, medium effect = 0.06, large effect = 0.14. The data from the Feeling Scale and Felt Arousal Scale were also represented in a circumplex model, which described the affective state with respect to activation (high and low) and valence (positive and negative). A Bonferroni post-hoc test was applied where appropriate. Significance was set at P≤0.05.

## Results

### Physiological responses to exercise

**Heart rate.** Mean HR traces for each protocol are shown in **Fig 1**. At the start of the protocols, immediately following the warm up, there were no significant differences in baseline HR responses between the protocols (*P* = 0.532*)*. There was a significant effect of protocol on

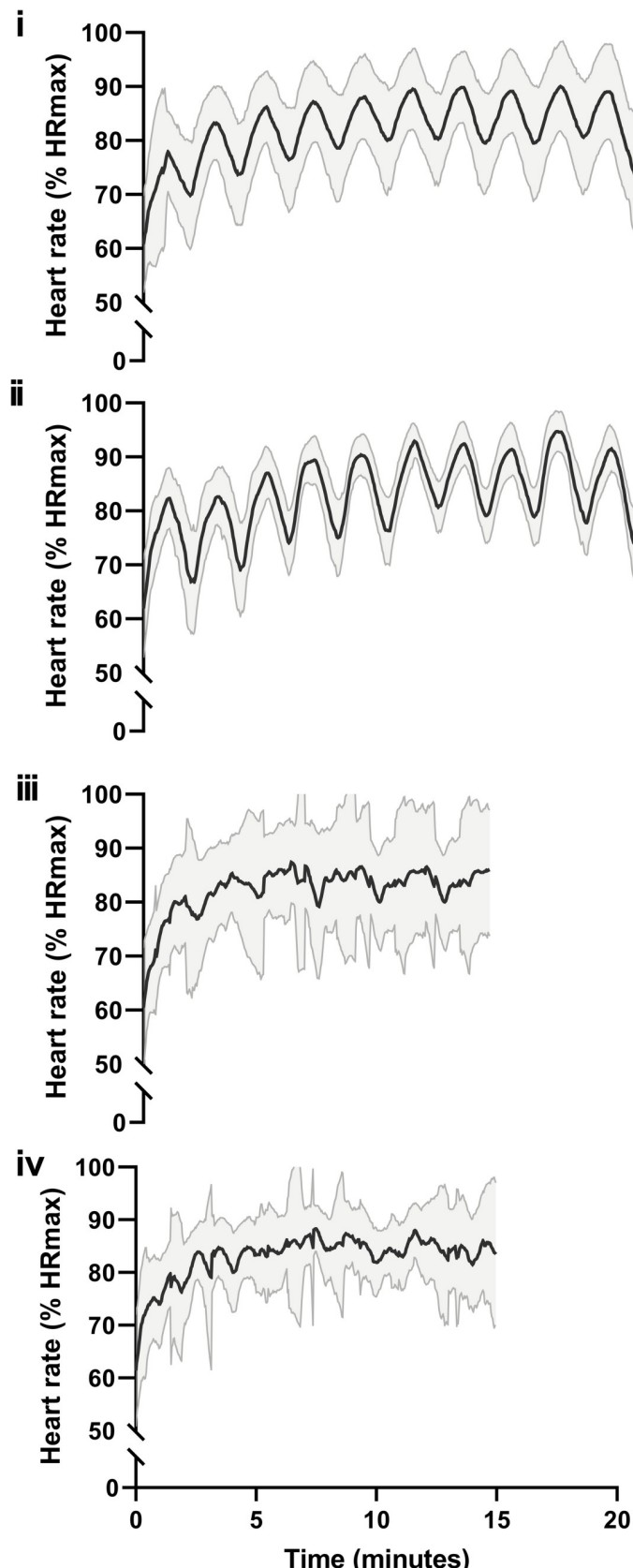

**Fig 1. Heart rate responses to the protocols.** Mean ± SD heart rate traces during **(i)** Ergometer-60:60 (Ergo-60:60; 10x1min with 1min recovery on a cycle ergometer), **(ii)** Body weight-60:60 (BW-60:60; 10x1min with 1min recovery, using whole-body exercises), **(iii)** Social Media-20:10 (SM-20:10; 5 sets of 4x20s with 10s rest.) and **(iv)** Social Media-40:20 (SM-40:20; 15x40s interval with 20s rest). Black solid line represents Mean and the grey shaded area the SD.

interval $HR_{peak}$ ($P = 0.018$, $\eta_p^2 = 0.238$), although following post-hoc analysis no significance was found between the HIIT protocols ($P>0.05$). Interval $HR_{mean}$ was not different between protocols ($P = 0.203$, $\eta_p^2 = 0.111$). There was also no difference between HIIT protocols for session $HR_{peak}$ ($P = 0.315$, $\eta_p^2 = 0.060$) or session $HR_{mean}$ ($P = 0.015$, $\eta_p^2 = 0.238$). There was a significant effect of protocol on the proportion of intervals meeting the ACSM high-intensity exercise criterion (HR >80% of maximum HR) ($P = 0.005$, $\eta_p^2 = 0.265$), with the criterion being achieved more regularly during BW-60:60 than SM-20:10 ($P = 0.039$), but no further differences observed during Ergo-60:60 or SM-40:20. There was also a significant effect of protocol on time spent above the criterion HR (HR >80% of max) ($P<0.001$, $\eta_p^2 = 0.488$), with participants spending significantly less time above 80% of $HR_{max}$ in SM-20:10 (8±3mins) than all other protocols (Ergo-60:60: $P = 0.034$, BW-60:60: $P = 0.006$, SM-40:20: $P = 0.006$), but no further differences between protocols. Data is presented in **Table 2**.

**Blood Lactate.**   There was no significant differences in baseline blood lactate between the protocols *(P = 0.218)*. Change in blood lactate was significantly different between protocols ($P<0.001$, $\eta_p^2 = 0.239$), SM-20:10 resulted in a significantly lower change in blood lactate concentration (5.5±2.6mmol/L) compared to Ergo-60:60 and BW-60:60 (7.4±2.6mmol/L and 7.7 ±3.7 mmol/L, $P = 0.002$ and $P<0.001$ respectively). There were no further differences between protocols ($P>0.05$; **Fig 2**).

## Perceptual responses during exercise

**Feeling scale.**   The minimum reported Feeling Scale score was similar across all protocols ($P = 0.268$, $\eta_p^2 = 0.051$; **Fig 3A**). Detailed information regarding Feeling Scale scores over time for each protocol is presented in **Fig 3B**, importantly markings on the figure represent *significant* changes compared to the following intervals. Feeling Scale scores immediately before the interval are also presented in **Fig 3B**. The Feeling Scale scores decreased in a linear manner after interval 5 during Ergo-60:60 and interval 6 during BW-60:60 (**Fig 3Bi** and **3Bii**. The response to SM-20:10 and SM-40:20 was more complex with large variations present (**Fig 3Biii** and **3Biv**).

**Table 2. Heart rate (HR) responses to the HIIT protocols.**

|  | Ergo-60:60 | BW-60:60 | SM-20:10 | SM-40:20 | P Value |
|---|---|---|---|---|---|
| Session $HR_{peak}$ (%) | 94±4 | 95±4 | 93±6 | 94±4 | P = 0.315 |
| Session $HR_{mean}$ (%) | 84±6 | 83±4 | 83±9 | 84±5 | P = 0.765 |
| Interval $HR_{peak}$ (%) | 90±5 | 90±3 | 84±7 | 87±4 | P = 0.018 |
| Interval $HR_{mean}$ (%) | 85±6 | 83±4 | 81±9 | 84±4 | P = 0.203 |
| HR ≥80% max (min) | 13.9±4.9* | 13.5±3.5* | 7.7±3.1 | 12.1±2.4* | P<0.001 |
| Proportion of intervals meeting a HR ≥80% max (%) | 87±16 | 93±10* | 74±20 | 88±15 | P = 0.005 |

Values are mean ± SD.

*Represents significant difference from SM-20:10 (P<0.05). Session $HR_{peak}$: Maximum heart rate achieved during the whole exercise session. Session $HR_{mean}$: Mean heart rate achieved during the whole exercise session. Interval $HR_{peak}$: Average maximum heart rate achieved during each of the intervals only. Interval $HR_{mean}$: Average mean heart rate achieved during each of the intervals only. HR ≥ 80% max: Time spent above or equal to the high-intensity criterion (80% of maximum heart rate). Proportion of intervals meeting a HR ≥ 80% max, proportion of the intervals meeting the high-intensity criterion (≥80% of maximum heart rate).

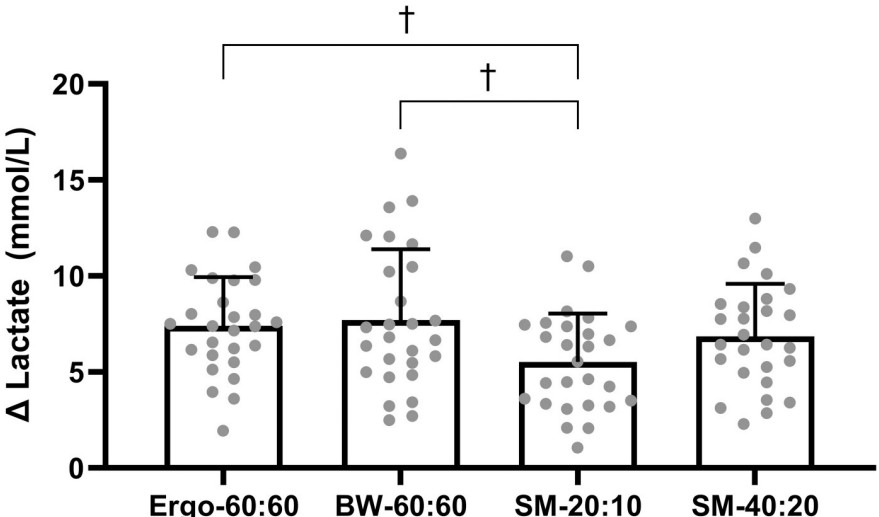

**Fig 2. Change in lactate during the HIT protocols.** † Represents significant difference from Social-Media-1 (SM-20:10) ($P < 0.05$).

**Circumplex model.** In order to investigate the nature and the magnitude of affect changes that occur in response to acute exercise stimuli a circumplex model was used. In a circumplex model of affect the horizontal axis represents affective valence (negative to positive) and the vertical axis represents the degree of perceived activation (low to high). Based on visual inspection, the patterning of Feeling Scale and Felt Arousal Scale values between Ergo-60:60 and BW-60:60 was similar within the circumplex model depicted in **Fig 4A and 4B**. During Ergo-60:60 and BW-60:60 the Feeling Scale shifted left toward greater displeasure after each interval, and Felt Arousal Scale shifted up towards a high arousal during the protocols, but only reached the activated pleasant or 'energy' quadrant after the 9th interval (out of a total of 10 intervals). SM-20:10 and SM-40:20 initially followed this pattern, however past interval 11 in SM-20:10 (out of a total of 20 intervals) and the 9th interval in SM-40:20 (out of a total of 15 intervals) the results fluctuate (**Fig 4C and 4D**). Unlike all other protocols, SM-20:10 remains in the unactivated pleasant or 'calmness' quadrant throughout the session.

## Motivational responses to exercise

**Intrinsic motivation inventory.** The subscale score for interest/enjoyment was significantly different between protocols ($P = 0.006$, $\eta_p^2 = 0.158$), BW-60:60 reported significantly higher scores ($5.0 \pm 1.2$) on the interest/enjoyment subscale compared to Ergo-60:60 ($4.4 \pm 1.2$, $P = 0.020$) and SM-40:20 ($4.3 \pm 1.2$, $P = 0.008$), with no other significant differences between the protocols ($P > 0.05$; **Fig 5A**). The subscale score for perceived competence was significantly different between protocols ($P < 0.001$, $\eta_p^2 = 0.226$), BW-60:60 and SM-20:10 reported significantly higher scores ($3.8 \pm 0.9$, $P = 0.005$ and $4.1 \pm 1.1$, $P = 0.001$ respectfully) on the perceived competence subscale compared to SM-40:20 ($3.2 \pm 1.1$), with no other significant differences between the protocols ($P > 0.05$; **Fig 5B**).

## Discussion

The main finding of the present study is that important acute physiological, perceptual and motivational differences exist between HIIT protocols developed for social media platforms and those shown to be effective in academic literature. In addition, our data suggests higher

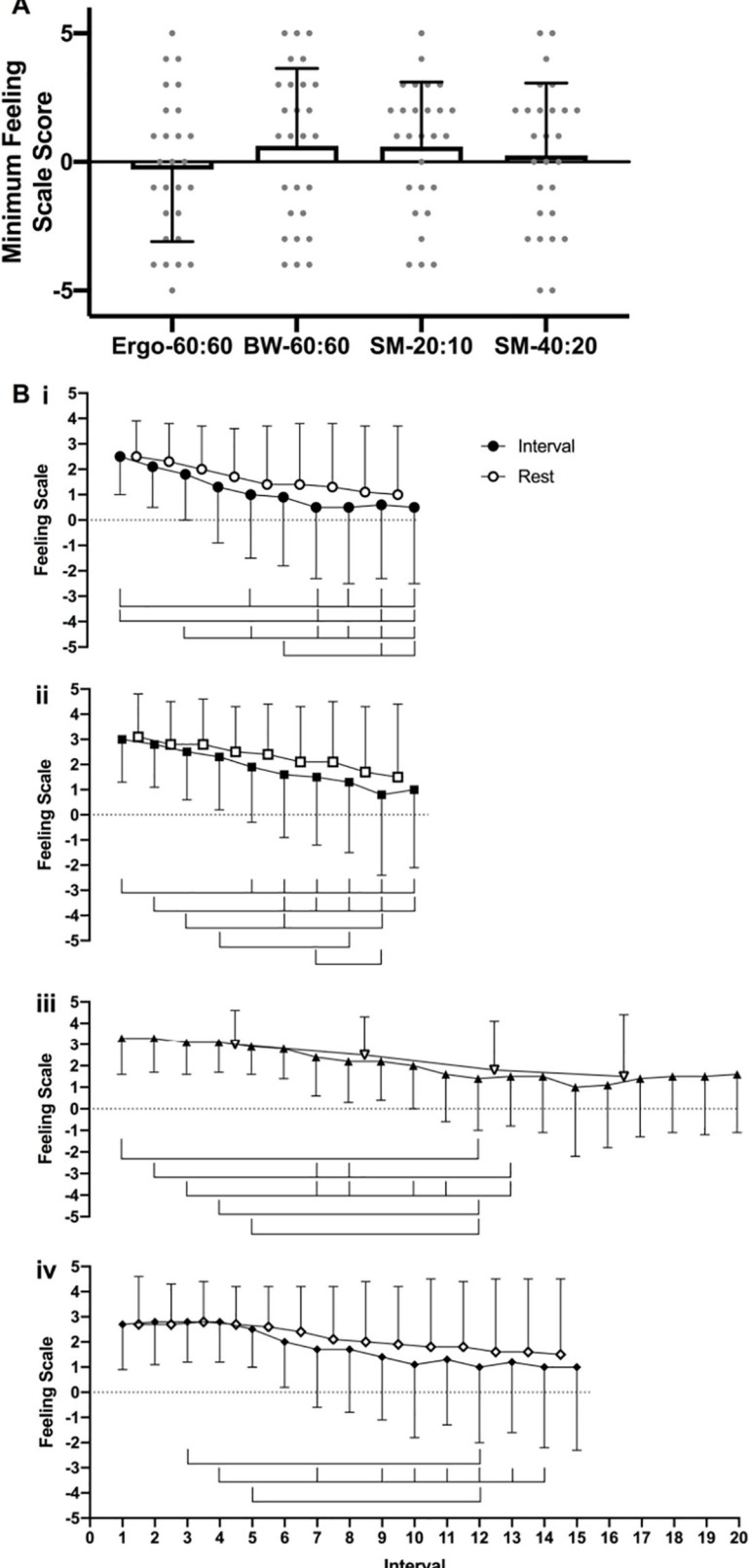

**Fig 3. Feeling scale (FS) responses to the protocols. A.** Minimum recorded Feeling Scale score **B.** Feeling Score over time during (i) Ergo-60:60, (ii) BW-60:60, (iii) SM-20:10 and (iv) SM-40:20. Closed icons represent FS recorded at the

end of each interval, open icons represent FS recorded at the end of the rest period. The markings above Fig 3B represent significance differences in Feeling Scale score immediately after the interval compared to the following intervals (P<0.05).

physiological responses experienced during HIIT are not a key determinant of post-exercise enjoyment or feelings of competence. Finally, in contrast to traditional HIIT protocols performed on a cycle ergometer, protocols performed using body-weight exercises result in more complex perceptual responses during exercise, which do not correlate with HR responses.

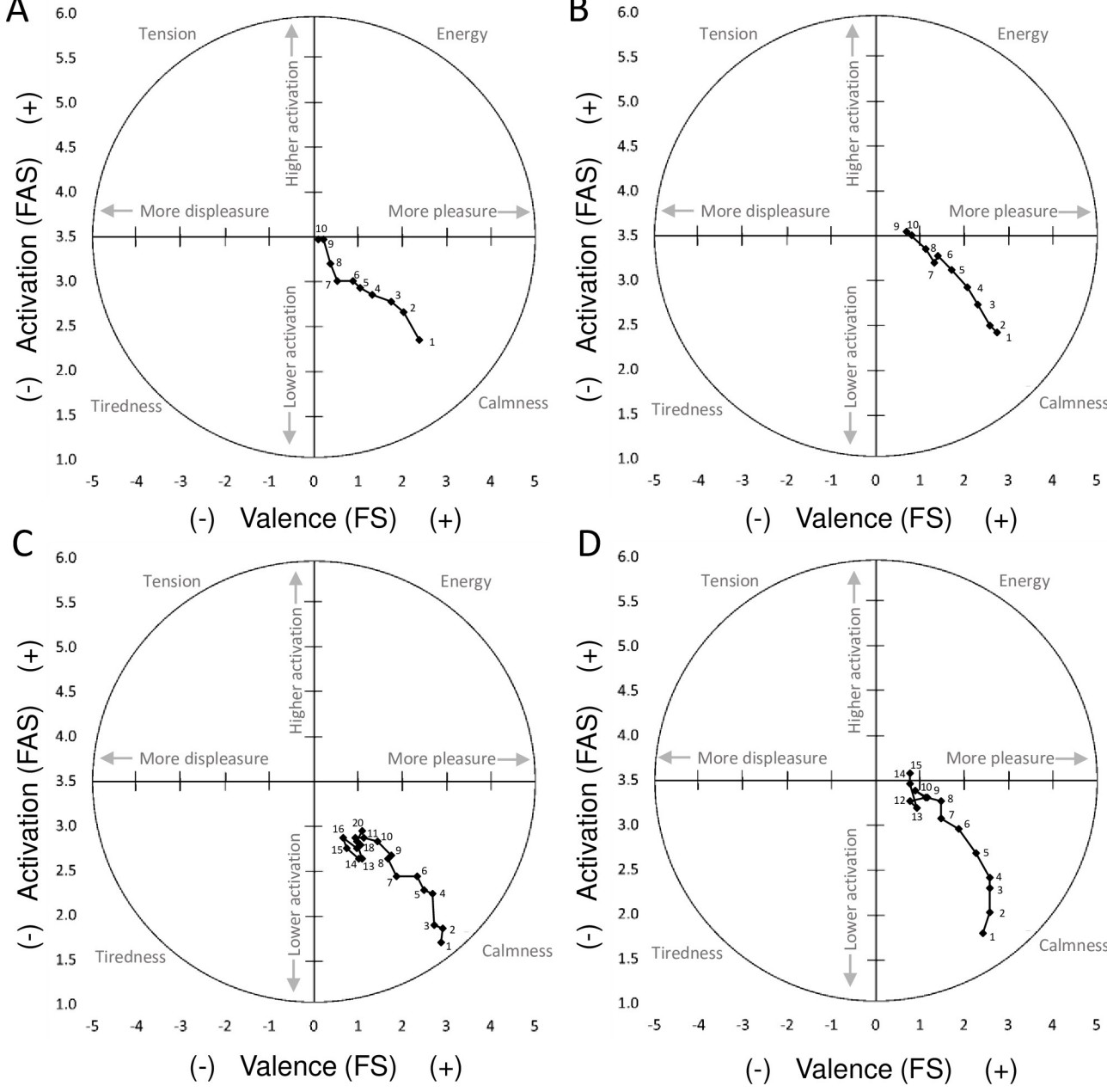

**Fig 4. Circumplex model to representing Feeling Scale (FS) and Felt Arousal Scale (FAS) responses to the protocols. A**. Ergo-60:60 **B.** BW-60:60 **C.** SM-20:10 **D.** SM-40:20. Values on the line represent the interval number when measurement was taken.

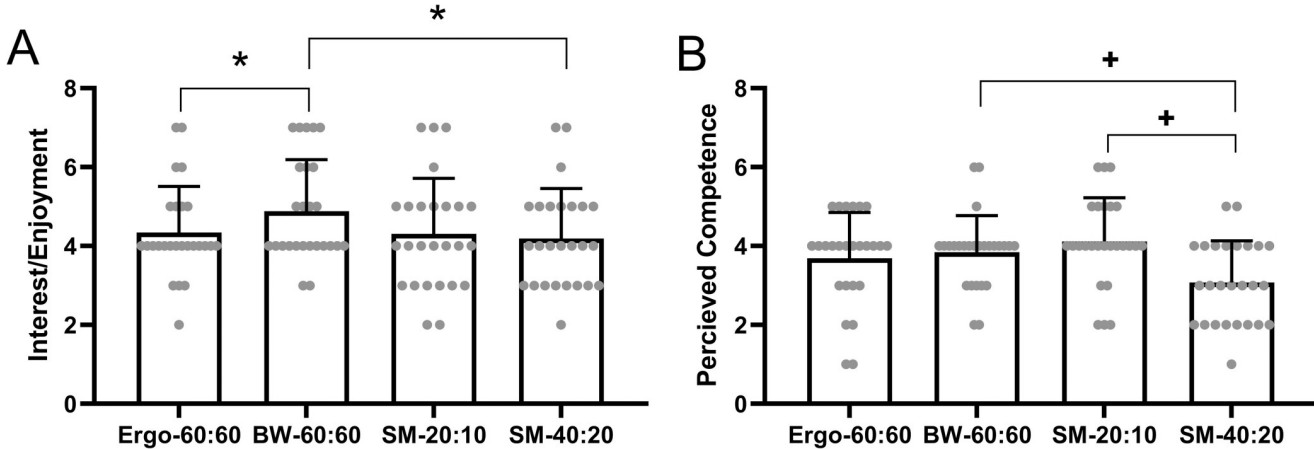

**Fig 5. Intrinsic motivation inventory responses to the HIT protocols.** * represents significant difference from BW-60:60 ($P < 0.05$). + represent significant difference to SM-40:20 ($P < 0.05$).

Therefore, application of traditional models (e.g. Dual-theory) may not be appropriate to describe the perceptual responses to body-weight HIIT. Future research should seek to investigate the physiological and perceptual responses to exercise throughout a body-weight HIIT intervention to determine long-term feasibility and effectiveness within a real world setting.

## Physiological responses to exercise

This is the first study to compare the physiological responses to popular HIIT workouts found on social media with established evidence based protocols [26,29]. Social media workouts are an attractive and popular alternative to traditional forms of HIIT; SM-20:10 and SM-40:20 have over 6 million views on YouTube, but research into their effectiveness is lacking. Recent work [8,9,34] suggests that acute physiological responses may dictate long-term training outcomes to HIIT. Therefore, a comparison of these social media protocols to those already established as effective in a research setting provides important information for consumers and fitness professionals.

Interestingly, similar HR and blood lactate responses were seen when comparing the two established protocols (Ergo-60:60 and BW-60:60), despite the different modalities of exercise (cycle ergometer vs body-weight). Furthermore, despite visible differences in the mean HR traces (**Fig 1**) there were no acute physiological differences between the established protocols and SM-40:20. In contrast a number of significant differences (lower change in lactate, less time spent above the criterion HR, and lower proportion of intervals meeting the criterion HR (>80 of max)) were observed between SM-20:10 and the established protocols. Moholdt et al. [34] reported greater increases in $VO_{2peak}$ in patients who achieved a higher $HR_{mean}$ during HIIT intervals (12 weeks, 4x4min at 85–95% HRmax with 3 mins rest at 60–70% HRmax), in patients with coronary heart disease. In addition, recent studies have suggested that lactate accumulation during HIIT is linked to the magnitude of the physiological adaptations. Hoshino et al. [9] administered mice with dichloroacetate (DCA), a pyruvate dehydrogenase activator which reduces muscle and blood lactate concentrations during and after exercise, over a 4 week HIIT period (10x60s high intensity treadmill running with a 1 min rest). Chronic DCA administration attenuated exercise-induced metabolic adaptations, including increases in mitochondrial enzyme activity (CS and b-HAD) and protein content (COXIV) compared to control animals (saline), suggesting that repeated lactate accumulation during HIIT is i

mportant for training–induced mitochondrial adaptations. Furthermore, Fiorenza et al. [8] found that speed endurance exercise (18x5s "all-out" efforts interspersed with 30s of passive recovery) increased PGC-1α mRNA response compared to work matched repeated-sprint exercise (6x20 s "all-out" with 120 s of passive recovery). Importantly, speed endurance exercise was associated with higher muscle lactate accumulation and lower muscle pH, suggesting that that greater metabolic perturbations with high lactate accumulation contributed to the enhanced PGC-1α mRNA response. As such, it is hypothesised that the lower time spent above the criterion HR (>80% of max) and change in lactate observed with SM-20:10 compared to the other protocols will reduce its long-term effectiveness. However, the data would suggest that body-weight exercises can be used as an effective HIIT modality, capable of eliciting similar acute physiological responses to HIIT performed on a laboratory cycle ergometer. Furthermore, protocols available via social media platforms can result in similar acute physiological responses, but fitness professionals need to proceed with caution when prescribing these protocols as they are not all equal. Interestingly, the lower lactate responses observed following SM-20:10 may have been due to the reduced interval duration as previous research in regional-level cyclists reported higher blood lactate responses after longer intervals (90s and 130s) compared to shorter 10s interval [35].

## Perceptual responses to exercise

Dual-Mode theory suggests affect experienced during exercise is influenced, in part, by the metabolic demand associated with the exercise [12]. However, in the current study the lowest recorded value on the feeling scale was not different between the protocols, despite significant differences in the physiological responses. This data contrasts with previous comparisons of HIIT protocols where findings have shown greater physiological strain is associated with lower affective responses [23,36], supporting the application of Dual-Mode theory for HIIT. Although the exercise intensity was different between HIIT protocols the same interval duration and work-to-rest ratios were employed in these earlier studies [23,36]. This contrasts with the current study where work-to-rest ratio and interval duration were different between the protocols. The potential importance of interval duration and work-to-rest ratio in determining affective response to HIIT is supported by recent research [37,38]. Martinez et al. [38] demonstrated that shorter intervals (30 and 60 seconds) had similar affective responses, but longer intervals were perceived as more aversive (120 seconds). Wood et al.[37], showed no difference in affect when comparing a HIIT and SIT protocol, despite significantly greater lactate accumulation experienced during SIT. Importantly, the work-to-rest ratio and interval duration used in the HIIT and SIT protocols were again different (60 second intervals and a 1:1 work-rest-ratio in HIIT; and 30 second intervals and a 1:3 work-to-rest ratio in SIT). As previously suggested by Jung et al. [39], these studies may suggest that work-to-rest ratio and interval duration could influence affective response to HIIT, and that manipulating these factors could interfere with the utility of Dual-Mode theory for HIIT.

It is also important to note that Ergo-60:60 imposed a fixed intensity (100% $W_{max}$) on participants, whereas the other protocols used all-out but self-paced intensities. The aim of the current research was not to compare imposed vs self-selected intensities, as different exercise modalities were used (cycling vs. body-weight exercises). However, Kellogg et al. [40] demonstrated that self-paced HIIT resulted in more negative affect (FS) than fixed intensity HIIT, when cycling was used (both protocols 8x60s work with 60s rest). Therefore, it is possible that the intensity regimes (imposed vs self-selected) could have influenced the perceptual responses observed in the current study.

As well as the magnitude of the peak negative or positive affect, Decker and Ekkekakis [41] suggests that the rate of change in affect occurring during the exercise is also important.

Previous studies employing cycling [37,42–44] or running [45,46] have consistently reported that affect becomes less positive during exercise in response to acute HIIT. This finding was echoed in a scoping review of the literature by Stork et al. [47], who noted that nearly all of the studies assessing in-task affect have shown a significant decline during HIIT. This profile is shown in both Ergo-60:60 and BW-60:60, where in-task affect shows a significant decline from interval 5 onwards. In contrast, SM-20:10 and SM-40:20 do not show a significant fall in affect from the first to last interval and changes in affect show no obvious pattern. The circumplex model, which incorporates affective valence and perceived activation to give a more complete view of affective responses during exercise [48], also highlights the difference in affect responses when using the two social media videos. It is unclear what is causing this difference between the protocols, however the social connection within social media HIIT (e.g. led by an influencer), and how the influencer interacts with the audience may have altered the enjoyment or perception of the unpleasant exercise [49]. Therefore, future studies should look to investigate the influence of exercise videos, interval duration, work-to-rest ratio and the use of body-weight exercises on in-task affect.

## Motivation and enjoyment

This is the first study to compare post-exercise enjoyment of HIIT protocols employing different exercise modalities. In their scoping review Stork et al [47] cautioned that people's experiences during one form of interval exercise may not be the same as another. Our data provides novel evidence supporting this argument, identifying that participants reported greater enjoyment when HIIT was performed using body-weight exercises (BW-60:60) compared to a cycle ergometer (Ergo-60:60). Importantly, BW-60:60 and Ergo-60:60 (matched for interval duration and work-to-rest ratio) produced similar HR traces and overall physiological responses, suggesting that the exercise mode could be an important factor in the differential enjoyment. Interestingly, BW-60:60 was also more enjoyable than SM-40:20, despite body-weight exercises being employed during both protocols. The greater enjoyment experienced could have been influenced by the lower competence for completing SM:40:20 compared to BW-60:60. It is possible that the specific exercises employed during BW-60:60 and SM-20:10 were responsible for the greater perceived competence following these protocols compared to SM-40:20. Unlike BW-60:60 and SM-20:10 which used entirely whole-body exercise, SM-40:20 employed a combination of whole-body and upper body exercise. Additionally the social media influencer may have had a part to play in creating perceptions of competence via the description of the exercises, encouragement provided and behaviour change techniques [50]. This observation may prove important to exercise professionals when designing HIIT protocols, as people are inherently drawn to engage in behaviours that they feel competent to carry out [51].

## Limitations

It is important to note that the current study was conducted in young recreationally active participants, with a relatively small sample size (n = 27). A younger population may find social media based approaches more acceptable and relevant than other populations, as such, we are unable to generalise our findings to older physically inactive individuals. We are also unable to generalise our findings to other HIIT protocols used within research or available on social media. All sessions were completed in a lab environment, rather than traditional environments used for body weight exercises (home, gym or local park). Finally, there is a diverse range of videos available on social media, and features unrelated to HIIT (e.g. likeability and relatedness of the influencer) may cause individual changes to perceptual responses. Investigating these factors was beyond the scope of the current study, but should be investigated in future work.

However, the work still represents an important step forward in our understanding of HIIT as it is the first study to explore the differences between established evidence-based protocols and workouts with millions of views on social media. Given the importance of social media influencers for impacting health [6] and the popularity of HIIT on social media it is important that future research continues to consider the potential effects of such protocols.

## Conclusions

This study shows that HIIT protocols available on social media offer an interesting real-world alternative for promoting exercise participation. However, the public and fitness professionals need to evaluate HIIT protocols promoted on social media with care, as not all will produce comparable acute physiological responses to evidence-based HIIT. In addition, the study demonstrates significant differences in the rate of change in affect between the social media protocols and those established within the literature. Future studies should look to investigate these differences further to explore if the social connection or interaction with the audience created by influencers may be responsible for the difference. Finally, the study also showed enjoyment of HIIT may be influenced by exercise mode, body-weight vs. Ergometer. Therefore, this study is an important first step in evaluating how HIIT protocols promoted by social media compare to evidence based protocols with research to support their efficacy to improve cardiorespiratory fitness. Future studies should continue to investigate these highly popular and practical HIIT protocols, including their long term effects on exercise adherence and health outcomes.

## Author Contributions

**Conceptualization:** Katie L. Hesketh, Florence Kinnafick, Anton J. M. Wagenmakers, Matthew Cocks, Juliette A. Strauss.

**Data curation:** Matthew Cocks, Juliette A. Strauss.

**Formal analysis:** Katie L. Hesketh, Florence Kinnafick, Matthew Cocks.

**Investigation:** Katie L. Hesketh.

**Methodology:** Katie L. Hesketh, Hannah Church, Florence Kinnafick, Sam O. Shepherd, Anton J. M. Wagenmakers, Matthew Cocks, Juliette A. Strauss.

**Project administration:** Katie L. Hesketh, Hannah Church.

**Supervision:** Florence Kinnafick, Sam O. Shepherd, Anton J. M. Wagenmakers, Matthew Cocks, Juliette A. Strauss.

**Writing – original draft:** Katie L. Hesketh, Matthew Cocks.

**Writing – review & editing:** Katie L. Hesketh, Hannah Church, Florence Kinnafick, Sam O. Shepherd, Anton J. M. Wagenmakers, Matthew Cocks, Juliette A. Strauss.

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
