## [Decision Letter · Decision Letter 0]

3 Jun 2021

PONE-D-21-08687

Evidence-based vs. social media based high-intensity interval training protocols: physiological and perceptual responses

PLOS ONE

Dear Dr. Cocks,

Thank you for submitting your manuscript to PLOS ONE. After careful consideration, we feel that it has merit but does not fully meet PLOS ONE’s publication criteria as it currently stands. Therefore, we invite you to submit a revised version of the manuscript that addresses the points raised during the review process.

Your manuscript has been reviewed by three experts in the field. As detailed below, they are generally supportive of your work, but have some queries and concerns about aspects of the methods and data interpretation and reporting. Please ensure you fully address these in your resubmission. Please also ensure to check all references for accuracy.  

We look forward to receiving your revised manuscript.

Kind regards,

Kathryn L. Weston, PhD

Academic Editor

PLOS ONE

Journal Requirements:

[Note: HTMLmarkup is below. Please do not edit.]

Reviewers' comments:

Reviewer's Responses to Questions

**Comments to the Author**

1. Is the manuscript technically sound, and do the data support the conclusions?

Reviewer #1: Yes

Reviewer #2: Yes

Reviewer #3: Yes

2. Has the statistical analysis been performed appropriately and rigorously? 

Reviewer #1: Yes

Reviewer #2: Yes

Reviewer #3: Yes

3. Have the authors made all data underlying the findings in their manuscript fully available?

Reviewer #1: Yes

Reviewer #2: Yes

Reviewer #3: Yes

4. Is the manuscript presented in an intelligible fashion and written in standard English?

Reviewer #1: Yes

Reviewer #2: Yes

Reviewer #3: Yes

5. Review Comments to the Author

Reviewer #1: Thank you for the opportunity to review this paper. This paper presents a randomised cross-over trial, exploring the acute physiological and perceptual responses to four high-intensity interval training protocols. I enjoyed reading this interesting paper, and feel it will be a novel contribution to the evidence base. I do however have a number of comments detailed below for the authors consideration.

Major comments

In this study, heart rate is used to explore exercise intensity. The authors present their findings as a percentage of maximal heart rate (HRmax). While this information is not included in the methods, it appears from information presented in Table 2 that the authors used predicted HRmax using the formula 220- participants age. However, as part of the first experimental visit, participants undertook an incremental exercise test until exhaustion, with heart rate monitored throughout. I wondered why, given the limitations associated with HRmax predictions, the authors chose to use this when they had access to heart rate data from the VO2max test, where presumably a maximum heart rate was recorded? I would have preferred to see the use of HRmax from the VO2max test, rather than predicted HRmax, if possible. If this data is not available, the use of predicted HRmax should be described in the methods, and should also be discussed as a limitation in the discussion.

Line 79-83: Here the authors begin to justify their selection of evidence-based and social media HIIT protocols, which I commend and think is central to the paper. I think this information might be better placed in the methods section (perhaps line 188 under the heading Training Protocols). It would also be useful to further explain why these protocols were chosen. Have the evidence-based protocols been shown to improve health/ fitness outcomes? Additionally, why specifically where the social media protocols/ videos selected?

In terms of the rest of the paper, my main comments centre around ensuring that the information presented is clear for an individual not involved in the project, or not overly familiar with HIIT research. I have provided more detail as to where I think this detail could be added below.

Minor comments

Abstract:

Perhaps the first line of the abstract could focus on the evidence base surrounding the effects of HIIT on health/ fitness, rather than the popularity of HIIT with social media influencers?

Line 23: could mean peak heart rates as a percentage of max heart rate for each protocol be included in the abstract?

Introduction

Line 33: a definition of HIIT would be useful in the first paragraph of the introduction.

Line 41: It may be useful to point out that some HIIT protocols may not be classed as HIIT given the traditional definition of HIIT. Or that it is unknown if these social media protocols are classed as HIIT. This might further justify the conduct of this study.

Line 66: in the section around Dual Mode Theory- it might be useful to point out that the majority of work which DMT is based on use continuous high intensity exercise- not HIIT.

Line 70: I believe Self Determination Theory needs capital letters. Additionally, I think a reference to Deci and Ryans work on SDT would be useful here.

Line 70: I wonder why specifically SDT is discussed here? While there are other theories of motivation, it might be useful to clarify that SDT has been used to predict/ explore/ enhance physical activity/ exercise behaviour? E.g., Teixeira et al., (2012) https://doi.org/10.1186/1479-5868-9-78

The novelty of this study could be made clearer in the introduction. Perhaps exploration of the literature exploring the acute effects of other HIIT protocols could be used to justify why it is important to explore acute effects of HIIT before prescribing it in an intervention?

Methods

Line 85- how were participants recruited?

Line 86- is there a reference that could be cited here for the definition of recreationally active?

Line 135- I am not sure if I am correct here, but do the YouTube videos used need to be attributed to specific YouTube channels/ do the names of the channels need to be stated in the manuscript?

Line 156- I would prefer to see slightly more detail on the methods undertaken to explore time spent at or above the criterion high-intensity heart rate here. Weston describes both per protocol and intention to treat analysis protocols, so it would be useful to state which method was undertaken.

I may have missed it, but when was the IMI administered?

Results

Rather than stating just the p-values in the text of the results section, it would be useful to also present means/ mean differences and standard deviations or confidence intervals, depending on the data. This would allow the authors to explore the clinical/ practical significance of the findings in more detail, rather than relying solely on statistical significance.

Line 201- I commend the authors for exploring the HR data using processes outlined by Weston et al., (2015). For the data exploring the proportion of high-intensity repetitions spent at or above the high intensity criterion (e.g. 80% HRmax), could the authors consider reporting these findings as outlined in Weston et al. 2015. For example (taken directly from the abstract of Weston et al., 2015):

“…the median (interquartile range) proportion of repetitions meeting the high-intensity criterion was 58% (42% to 68%).”

Figure 1 is very useful to visualise the variation of data around the mean. For readers who are unfamiliar with this type of figure, could the authors consider stating in the key that the grey shading is the SD and black line is the mean HR?

Line 243: It would be useful here to clarify for the reader how many intervals were completed in each protocol, to avoid them having to return to the methods. E.g. after the 9th interval (out of a total of X number of intervals).

Discussion

It would be useful when interpreting the results to explore the practical or clinical meaningfulness of the findings rather than relying on statistical significance. For example, in Figure 5, the difference in perceived competence between groups appears to be about 0.5 to 1 point, while this is statistically significant, is it practically or clinically meaningful? Are participants likely to notice this difference? Is there a minimum clinically important difference for this scale that could be explored?

Line 284- what is the definition of a considerable difference? Have the authors defined this previously?

Line 285- is Figure 2 the correct figure to be referring to here?

Line 287- an overview of what these differences were would be useful here.

Line 287-293 seems to be repetition from the introduction. Could this section be summarised more briefly given that these papers are discussed in the introduction?

Line 309- what does mimic acute physiological responses mean?

Line 334- The use of DMT for HIIT has been critiqued previously (See Batterhams argument in Biddle and Batterham 2015 https://doi.org/10.1186/s12966-015-0254-9). Most dual mode studies are conducted using continuous high intensity exercise, not interval exercise. Your findings seem to support the notion that affective responses could be different for interval exercise, despite the intensity. Or Jung et al., 2016 may be useful doi: 10.3389/fpsyg.2015.01999

Line 348: The authors state that the research team gave no encouragement to participants during the intervals apart from providing advice on correct technique. I would like to understand why this decision was made? In the videos used for the social media HIIT protocols, the facilitators provide generalised words of encouragement and some level of human interaction. I realise this could not have been completely standardised across the participants for the evidence based protocols, but perhaps the authors could consider whether this human interaction and encouragement in the social media videos may have impacted on enjoyment in comparison to no encouragement at all in the evidence based protocols.

Line 352: Should this section be named motivation or enjoyment? It is named motivation but seems to discuss enjoyment more.

Line 374: Could the authors consider adding in that the findings cannot be applied to other HIIT protocols or modalities?

Line 385- research led HIIT or evidence-based HIIT- would be useful to be consistent throughout the paper.

Line 392- I think this sentence may need further clarity. How do the findings show how HIIT can be used to promote exercise?

Reviewer #2: PLOS ONE-d-08687

Evidence-based vs. social media based high-intensity interval training protocols:

physiological and perceptual responses

General comments: I was excited to read this work as I have conducted some research in this area and am always eager to read what others lab are doing in this area. This study is well-rationalized, follows proper methods, and the presentation of the Results and subsequent explanation are sound. Findings will be of interests to scientists and clinicians who use interval exercise in their facilities.

Specific comments: Please respond to the comments listed below regarding your paper—thank you.

Abstract—this is well written, yet I have one comment to make in line 28. You do not present HR data so how can you conclude that these social media based protocols are feasible? Only if HR attains 85 %HRmax are these protocols truly eliciting intensities equivalent to lab based HIIT?

Introduction—so the last line of this section is not entirely true; please see work from our laboratory exploring acute responses to a social media protocol and infuse these findings into your text here as this is not as novel of a topic with this citation included. https://pubmed.ncbi.nlm.nih.gov/28658082/

Methods—this is not a criticism but more a question—these protocols are not matched for work and have different structure, duration, etc., so how does this alter the interpretation of these data, as clearly the differences in these traits alter the magnitude of physiological and perceptual stress experienced?

Line 179: Please confirm that this was a two-way ANOVA comparing differences in these variables across time as well as bout; thank you.

Line 105: the 10 X 1 cycling protocol is prescribed according to Wmax-PPO, yet there is no text here denoting how this was done. Also, there is no mention of text in this section describing the fed state of participants pre-session, if time of day was standardized, if PA was prohibited prior to testing, etc.?

Were any practice sessions allotted to the participants to improve their familiarity with these body weight exercises?

Were the instructions on how to interpret FS standardized and was the same experimenter tasked with recording this outcome in each session?

I recommend that the Authors present some type of ES value in their Results to denote the meaningfulness of any differences—thank you.

Results—line 199—is there a reason why predicted HRmax is used here when your baseline VO2max test allows you to actually assess true HRmax? Please clarify this.

Line 219—I believe this text needs some additional p values to better articulate the statistical results; thank you.

Discussion—Lines 287-301 are nice but in my opinion, too replicative of the Introduction and in some ways, too speculative too. I think it would be best to condense some of this text and comment more on if the 20-10 bout (having the lowest interval duration and time > 80 %HRmax) is feasible and indicative of HIIE exercise vs. the other 3 regimens used.

Also I believe that some of this text needs to be substituted by data from similarly habitually active participants rather than mice or trained cyclists, who have different exercise tolerance, BLa accumulation, etc. https://pubmed.ncbi.nlm.nih.gov/28737586/

I also believe you need to talk about the fact that the 10 X 1 regimen is at a fixed intensity; whereas, the other protocols are all-out or self-paced. Thus, the first regimen is imposed upon each participant; whereas, in the other 3 sessions, the exerciser has total control of his/her effort exerted. There is work showing that this feature can alter perceptions of exercise, so perhaps a few lines of text needs to be included here acknowledging this attribute.

Reviewer #3: This is an interesting study examining acute physiological, perceptual and motivational responses to popular social media HIIT protocols in comparison to evidence-based HIIT protocols. I commend the researchers for their novel study, which is particularly timely given many people’s time at home has been significant during the past year and interest in social media based workouts has also increased.

The manuscript is very well written, with a strong and balanced discussion including key studies in this field and highlighting opportunities for future research.

You may wish to consider the points below:

Methods:

It would be useful to include further details regarding the four HIIT protocols. For example, where were the social media HIIT sessions completed? In the lab? Details of a warm-up were provided, however did participants also complete a cool-down?

In addition to the popularity of the YouTube clips, what considerations were made when choosing these two HIIT workouts?

Discussion:

It might be useful to consider the venue in which HIIT sessions were conducted when explaining findings. Enjoyment and motivation may differ for a lab based session in comparison to other venues (e.g. home, gym, outdoors, etc.). In addition, the variety of exercises included for the social media and BW HIIT protocols, in comparison to using only the cycle ergometer, may also explain differences in enjoyment and motivation. The age of participants might also be considered, as younger adults may find social media based PA approaches more acceptable and relevant than other age groups.

Limitations:

Participants being classified as recreationally active has been noted as a limitation of the study, however the sample size has not been mentioned.

6. PLOS authors have the option to publish the peer review history of their article (what does this mean?). If published, this will include your full peer review and any attached files.

Reviewer #1: No

Reviewer #2: No

Reviewer #3: No

---

## [Author Response · Author response to Decision Letter 0]

11 Aug 2021

We thank all of the reviewers for their valuable time and feedback. The comments provide have all been addressed. Any changes have been highlighted in red in the text and summarised below.

Reviewer #1: Thank you for the opportunity to review this paper. This paper presents a randomised cross-over trial, exploring the acute physiological and perceptual responses to four high-intensity interval training protocols. I enjoyed reading this interesting paper, and feel it will be a novel contribution to the evidence base. I do however have a number of comments detailed below for the authors consideration.

Major comments

In this study, heart rate is used to explore exercise intensity. The authors present their findings as a percentage of maximal heart rate (HRmax). While this information is not included in the methods it appears from information presented in Table 2 that the authors used predicted HRmax using the formula 220- participants age. However, as part of the first experimental visit, participants undertook an incremental exercise test until exhaustion, with heart rate monitored throughout. I wondered why, given the limitations associated with HRmax predictions, the authors chose to use this when they had access to heart rate data from the VO2max test, where presumably a maximum heart rate was recorded? I would have preferred to see the use of HRmax from the VO2max test, rather than predicted HRmax, if possible. If this data is not available, the use of predicted HRmax should be described in the methods, and should also be discussed as a limitation in the discussion.

Originally the manuscript presented data as a percentage of predicted HRmax (220-age) to replicate the real world application of using social media HIIT (i.e. home-based exercise in those without access to formal physiological testing). However, the authors agree that in this context the HRmax acquired from the VO2max test would result in greater insight into the acute responses. Figure 1 and Table 2 have been updated to reflect these changes, and statical analysis has been reproduced. Following these changes, post-hoc analysis did not show a significant difference between protocols when HR was expressed as interval HRpeak (line 233). Significance remained the same for time spent >80%HRmax and proportion of intervals meeting a HR ≥80% max. This change to the results has resulted in some minor changes to the text in the discussion, but we do not believe that the conclusions drawn have been affected, as such, these have not been amended. 

Line 79-83: Here the authors begin to justify their selection of evidence-based and social media HIIT protocols, which I commend and think is central to the paper. I think this information might be better placed in the methods section (perhaps line 188 under the heading Training Protocols). It would also be useful to further explain why these protocols were chosen. Have the evidence-based protocols been shown to improve health/ fitness outcomes? Additionally, why specifically where the social media protocols/ videos selected?

We thank the reviewer for this suggestion and as suggested have moved this information to the methods (line 137). In addition, the new paragraph within the methods under the heading of ‘Training Protocols’ now provides greater insight into why the protocols were selected. In brief, we used 4 criteria to assess videos found on YouTube 1) had to be featured on a popular YouTube fitness channel 2) have ‘HIIT’ in the title of the video 3) take less than 20 minutes, to take advantage of the time-saving nature of HIIT 4) include body weight exercises with no equipment. From these videos the SM-20:10 protocol was included as the video uses “Tabata training”, a variation of the original protocol designed by Tabata et al. (1996) which has been demonstrated to lead to increases in VO2peak. SM-40:20 was included as the protocol used a blend of aerobic and resistance-based exercises (e.g. press-ups) which would not typically fall under the traditional definition of HIIT, but is used by a number of videos found on social media channels.

In terms of the rest of the paper, my main comments centre around ensuring that the information presented is clear for an individual not involved in the project, or not overly familiar with HIIT research. I have provided more detail as to where I think this detail could be added below.

Minor comments

Abstract:

Perhaps the first line of the abstract could focus on the evidence base surrounding the effects of HIIT on health/ fitness, rather than the popularity of HIIT with social media influencers?

Line 11 within the abstract now reads “High intensity interval training (HIIT) is a time-efficient exercise modality to improve cardiorespiratory fitness, and has been popularised by social media influencers.” We hope this presents the evidence base for HIIT but also refers to the role of social media influencers, which, we believe is important in the context of our work. 

Line 23: could mean peak heart rates as a percentage of max heart rate for each protocol be included in the abstract?

Due to the updated analysis requested by the reviewers (the use of actual HRmax used rather than predicted HRmax), percentage of intervals achieving a heart rate greater than 80% HRmax has now been used within the abstract. The values for BW-60:60 and SM-20:10 have been added (Line 25), however adding any additional data within the abstract would exceed the abstract word count and would result in an unclear message for an abstract. The abstract now includes all data surrounding the time spent above 80%HRmax for each protocol (Line 28). 

Introduction

Line 33: a definition of HIIT would be useful in the first paragraph of the introduction.

A definition has been added to the introduction (line 38) “High intensity interval training includes brief, intermittent bursts of vigorous activity (typically between 80-100% HRmax), interspersed by periods of rest or recovery (1).” 

Line 41: It may be useful to point out that some HIIT protocols may not be classed as HIIT given the traditional definition of HIIT. Or that it is unknown if these social media protocols are classed as HIIT. This might further justify the conduct of this study.

Thank you for your comment we agree that our rationale for investigating social media HIIT protocols could be better developed within this paragraph. Therefore, we have added further information here (line 57), including the observation that many social media protocols do not use interval durations and/or work-to rest ratios that have been backed by research. In addition, many social media protocols use a combination of aerobic and resistance exercises which we do not believe meet the research backed definition of HIIT. 

Line 66: in the section around Dual Mode Theory- it might be useful to point out that the majority of work which DMT is based on use continuous high intensity exercise- not HIIT.

The following statement has been added to the introduction (Line 83): “However, the majority of work used to support Dual Mode Theory has been carried out using continuous high intensity exercise, and its use within HIIT has been critiqued previously (15).” 

Line 70: I believe Self Determination Theory needs capital letters. Additionally, I think a reference to Deci and Ryans work on SDT would be useful here.

Thank you - SDT has been capitalised (Line 87), and the citation for Deci and Ryan (2000) has been added to Line 89. 

Line 70: I wonder why specifically SDT is discussed here? While there are other theories of motivation, it might be useful to clarify that SDT has been used to predict/ explore/ enhance physical activity/ exercise behaviour? E.g., Teixeira et al., (2012) https://hes32-ctp.trendmicro.com:443/wis/clicktime/v1/query?url=https%3a%2f%2fdoi.org%2f10.1186%2f1479%2d5868%2d9%2d78&umid=2ba797c4-0763-42b4-9ef0-d2ada0e77048&auth=768f192bba830b801fed4f40fb360f4d1374fa7c-d09700cffeed5beae5d926dd1a8b45900ac6a33e

We agree the link could have been made clearer within this section as it reflects the potential for HIIT protocols with positive changes to perceived competence and autonomy to influence future exercise participation. We have therefore edited this section (Line 85-90) to improve clarity of the argument. 

The novelty of this study could be made clearer in the introduction. Perhaps exploration of the literature exploring the acute effects of other HIIT protocols could be used to justify why it is important to explore acute effects of HIIT before prescribing it in an intervention?

As far as we are aware, several studies have investigated people’s attitudes and intentions toward interval exercise, but none have tested if these variables can predict subsequent interval exercise behaviour. Therefore the acute responses to affect are only theory-driven, although we agree that future theory-driven research is needed to address questions regarding perceptual responses, interval exercise, and long-term behaviour.

Methods

Line 85- how were participants recruited?

Participants were recruited from Liverpool John Moores University via internal emails and posters. This information has been added to Line 101.

Line 86- is there a reference that could be cited here for the definition of recreationally active?

To our knowledge there is no accepted definition of recreationally active, but in our study it was greater than 1 but less than 4 structured exercise sessions per week. This is similar to what was used in previous studies (e.g. https://www.ncbi.nlm.nih.gov/pmc/articles/PMC3761819/). As this is not a formal definition we have not added a reference here. 

Line 135- I am not sure if I am correct here, but do the YouTube videos used need to be attributed to specific YouTube channels/ do the names of the channels need to be stated in the manuscript?

We have added which YouTube channel the HIIT videos were featured on to Line 172 and Line 179. 

Line 156- I would prefer to see slightly more detail on the methods undertaken to explore time spent at or above the criterion high-intensity heart rate here. Weston describes both per protocol and intention to treat analysis protocols, so it would be useful to state which method was undertaken.

When addressing the reviewers comments on the manuscript we realised that there has been an amendment to Westerns et al. The reference should now be cited as Taylor et al, this has been amended in the text. 

The authors interpretation of the guidelines presented in Taylor et al. (2015) are that these are for use within intervention studies, where assessment of session attendance and compliance with the prescribed protocol are crucial. As such, in Taylor et al. (2015) the intention to treat analysis includes analysis of training sessions regardless of if they were completed or not (a value of 40% of maximal heart rate was imputed for cases where heart rate data were missing due to participant absence), whereas, per protocol analysis included only sessions that were completed. The use of intention to treat analysis in this context is crucial to quantify the overall dose of the intervention, whereas the per protocol analysis provides information on intervention fidelity. As our study is an acute assessment of 4 HIIT protocols we do not believe that the use of intention to treat or per protocol analyses is warranted or intended by Taylor et al. We believe that this difference is rather complex and as such the current manuscript would not benefit from discussion of the different statistical approaches taken. 

I may have missed it, but when was the IMI administered?

This has been added to Line 135 for clarity “10 minutes after completion of the protocol”

Results

Rather than stating just the p-values in the text of the results section, it would be useful to also present means/ mean differences and standard deviations or confidence intervals, depending on the data. This would allow the authors to explore the clinical/ practical significance of the findings in more detail, rather than relying solely on statistical significance.

For the heart rate data, we feel as though these values are better represented within the table as the clarity of the data interpretation was compromised when all data was included within the text. However following the suggestions from reviewer 2 we have added means to the results section when they are presented as a figure, and have also added effect sizes to the results, which allows for meaningfulness of any differences to be interpreted. 

Line 201- I commend the authors for exploring the HR data using processes outlined by Weston et al., (2015). For the data exploring the proportion of high-intensity repetitions spent at or above the high intensity criterion (e.g. 80% HRmax), could the authors consider reporting these findings as outlined in Weston et al. 2015. For example (taken directly from the abstract of Weston et al., 2015):

“…the median (interquartile range) proportion of repetitions meeting the high-intensity criterion was 58% (42% to 68%).”

Our aim was to use the guidance provided by Taylor et al (2015) as this the only paper to provide recommendations on reporting HIIT interventions. However as mentioned in our previous comment the aim of Taylor et al (2015) was to provide intervention fidelity. We believe that the median was used in Taylor et al due to the lack of data recorded during the intervention, as such they replaced missing HR values with 40%. However our data is from 4 acute session therefore we believe that presenting the mean±SD is more relevant within the context of our work. 

Figure 1 is very useful to visualise the variation of data around the mean. For readers who are unfamiliar with this type of figure, could the authors consider stating in the key that the grey shading is the SD and black line is the mean HR?

Added to the legend of Figure 1“Black solid line represents Mean and the grey shaded area the SD.”

Line 243: It would be useful here to clarify for the reader how many intervals were completed in each protocol, to avoid them having to return to the methods. E.g. after the 9th interval (out of a total of X number of intervals).

This additional detail has been added for each of the protocols (Line285-287).

Discussion

It would be useful when interpreting the results to explore the practical or clinical meaningfulness of the findings rather than relying on statistical significance. For example, in Figure 5, the difference in perceived competence between groups appears to be about 0.5 to 1 point, while this is statistically significant, is it practically or clinically meaningful? Are participants likely to notice this difference? Is there a minimum clinically important difference for this scale that could be explored?

To our knowledge there isn’t a clinical significance for these values as they are dependent on each individual. Although we have now included effect sizes within the results section to indicate the meaningfulness of any differences, which previous work investigating acute responses to HIIT have also included. We agree that future studies are needed to investigate the effect perceptual variables have on adherence, and to provide reference values as a result. 

Line 284- what is the definition of a considerable difference? Have the authors defined this previously?

This has been edited to ‘visual differences’ rather than ‘considerable difference’, as we agree the phrase ‘considerable’ is not clear/descriptive of the data presented. The aim of this sentence was to bring attention to the visual differences in the heart rate traces during the different protocols. 

Line 285- is Figure 2 the correct figure to be referring to here?

Thank you for pointing this out, updated to Figure 1

Line 287- an overview of what these differences were would be useful here.

We have added information that lower change in lactate, less time spent above 80% HRmax and a lower proportion of intervals spent above 80%HRmax were the differences observed (Line 333). - 

Line 287-293 seems to be repetition from the introduction. Could this section be summarised more briefly given that these papers are discussed in the introduction?

We feel this section is more impactful and clearer has part of the development of the discussion. But agree it was repetitive of the text used in the introduction, therefore we have edited the introduction to summarise the literature to prevent duplication (Line 62). 

Line 309- what does mimic acute physiological responses mean?

We agree that the choice of wording could have been clearer, this has been changed to “result in similar” to improve clarity (Line 358). 

Line 334- The use of DMT for HIIT has been critiqued previously (See Batterhams argument in Biddle and Batterham 2015 https://hes32-ctp.trendmicro.com:443/wis/clicktime/v1/query?url=https%3a%2f%2fdoi.org%2f10.1186%2fs12966%2d015%2d0254%2d9&umid=2ba797c4-0763-42b4-9ef0-d2ada0e77048&auth=768f192bba830b801fed4f40fb360f4d1374fa7c-03e42631fbed81d32922dee497184693e18d7952). Most dual mode studies are conducted using continuous high intensity exercise, not interval exercise. Your findings seem to support the notion that affective responses could be different for interval exercise, despite the intensity. Or Jung et al., 2016 may be useful doi: 10.3389/fpsyg.2015.01999

Thank you, this has been added to line 381. 

Line 348: The authors state that the research team gave no encouragement to participants during the intervals apart from providing advice on correct technique. I would like to understand why this decision was made? In the videos used for the social media HIIT protocols, the facilitators provide generalised words of encouragement and some level of human interaction. I realise this could not have been completely standardised across the participants for the evidence based protocols, but perhaps the authors could consider whether this human interaction and encouragement in the social media videos may have impacted on enjoyment in comparison to no encouragement at all in the evidence based protocols.

We did not provide encouragement to participants during the protocols as we wanted to replicate the home-based environment that these protocols would be used in. There are no videos currently available for the evidence based protocols, as such, if these were completed by participants at home they would have no encouragement. We believe that we have commented on the potential role of human interaction in the social media protocols in lines 424 of the discussion, but have added more information to line 438 in the limitations. 

Line 352: Should this section be named motivation or enjoyment? It is named motivation but seems to discuss enjoyment more.

We agree that this sections primarily discusses enjoyment, however it also presents results from the IMI which includes perceived competence. We have therefore updated the section to ‘Motivation and Enjoyment’ to more accurately reflect the content (Line 209)

Line 374: Could the authors consider adding in that the findings cannot be applied to other HIIT protocols or modalities?

This had been added to the limitations (line 434)

Line 385- research led HIIT or evidence-based HIIT- would be useful to be consistent throughout the paper.

Changed to evidence based throughout 

Line 392- I think this sentence may need further clarity. How do the findings show how HIIT can be used to promote exercise?

We agree that this statement was rather vague and have changed it to read, “this study is an important first step in evaluating how HIIT protocols promoted by social media compare to evidence based protocols with evidence to support their efficacy to improve cardiorespiratory fitness” (Line 457). 

Reviewer #2: PLOS ONE-d-08687

Evidence-based vs. social media based high-intensity interval training protocols:

physiological and perceptual responses

General comments: I was excited to read this work as I have conducted some research in this area and am always eager to read what others lab are doing in this area. This study is well-rationalized, follows proper methods, and the presentation of the Results and subsequent explanation are sound. Findings will be of interests to scientists and clinicians who use interval exercise in their facilities. Specific comments: Please respond to the comments listed below regarding your paper—thank you.

Thank you for your review and valuable feedback, the comments provide have all been addressed. Any changes have been highlighted in red in the text and summarised below. 

Abstract—this is well written, yet I have one comment to make in line 28. You do not present HR data so how can you conclude that these social media based protocols are feasible? Only if HR attains 85 %HRmax are these protocols truly eliciting intensities equivalent to lab based HIIT?

Due to the updated analysis requested by the reviewers (the use of actual HRmax rather than predicted HRmax), percentage of intervals achieving a heart rate greater than 80% HRmax has now been used within the abstract. The response to each protocol has been added to the abstract (Line 25).

Introduction—so the last line of this section is not entirely true; please see work from our laboratory exploring acute responses to a social media protocol and infuse these findings into your text here as this is not as novel of a topic with this citation included. https://hes32-ctp.trendmicro.com:443/wis/clicktime/v1/query?url=https%3a%2f%2fpubmed.ncbi.nlm.nih.gov%2f28658082%2f&umid=2ba797c4-0763-42b4-9ef0-d2ada0e77048&auth=768f192bba830b801fed4f40fb360f4d1374fa7c-cc656fb46dcae2c1a350e96ae85fa406b896975c

We thank the reviewer for drawing attention to this previous work and have added reference to it in the introduction (line 52). However, although an important contribution to the literature we don’t feel this can be applied in the same context as the current study. Even though this study is based on a popular app, it is not a social media video (such as featured on YouTube), led by an unqualified social media influencer. Furthermore the paper compares this body weight protocol to a work matched protocol conducted on a cycle ergometer. This protocol (12x30s with 10s rest) has not previously been used within the literature (there is no evidence to show that it improves cardiorespiratory fitness). Therefore, we feel the statement “there is no research comparing the protocols used in these social media videos to those employed within the research.” is valid. 

Methods—this is not a criticism but more a question—these protocols are not matched for work and have different structure, duration, etc., so how does this alter the interpretation of these data, as clearly the differences in these traits alter the magnitude of physiological and perceptual stress experienced?

We agree and as outlined in the discussion believe that the different interval durations and work to rest ratios contribute significantly to the physiological and perceptual differences observed within the current study. We believe that future studies should continue to use the acute approach taken with the current study to investigate specifically how manipulating these different variables can influence physiological and perceptual outcomes, hopefully developing a protocol with high physiological load but also positive perceptual responses. 

Line 179: Please confirm that this was a two-way ANOVA comparing differences in these variables across time as well as bout; thank you.

We agree this was previously unclear, a one-way within subject ANOVA was conducted to investigate heart rate responses, change in lactate, lowest reported feeling scale score and responses to the IMI questionnaire, as these are not pre to post values. This has been updated within the text for clarity (Line 217). 

Line 105: the 10 X 1 cycling protocol is prescribed according to Wmax-PPO, yet there is no text here denoting how this was done. Also, there is no mention of text in this section describing the fed state of participants pre-session, if time of day was standardized, if PA was prohibited prior to testing, etc.?

The incremental exercise test was used to calculate Wmax (PPO), this detail can be found on Line 116. More detailed information has been added to Line 110 indicating that all visits were completed at a similar time of day and all participants were asked to refrain from vigorous exercise 24 hours before and to fast 3hrs before completing the protocol. 

Were any practice sessions allotted to the participants to improve their familiarity with these body weight exercises?

No all participants arrived to the lab without any prior knowledge of the exercise protocol/exercise they were going to perform. This information has been added to line 113 in the methods. 

Were the instructions on how to interpret FS standardized and was the same experimenter tasked with recording this outcome in each session?

Instructions were standardised and same member of the experimental team provided the instruction. This detail has been added to Line 201.

I recommend that the Authors present some type of ES value in their Results to denote the meaningfulness of any differences—thank you.

Thank you for this recommendation, we have added effect size values to the results to signify meaningful difference and improve interpretation of the data by the reader. Details of the approach taken has been added to the statistical analysis section, line 222. 

Results—line 199—is there a reason why predicted HRmax is used here when your baseline VO2max test allows you to actually assess true HRmax? Please clarify this.

Following the feedback from both reviewer 1 and yourself, we have now used the HRmax collected during the VO2max test. All results involving HR have now been updated to reflect a percentage of actual HRmax rather than predicted, and the appropriate statistical analysis has been recreated. Please see our response to reviewer one for further details. 

Line 219—I believe this text needs some additional p values to better articulate the statistical results; thank you.

Due to the complexities of the protocols and the number of intervals included within each protocol, we feel the results section would not benefit from additional P values here. All significant P values have been indicated on Figure 3 for the reader. 

Discussion—Lines 287-301 are nice but in my opinion, too replicative of the Introduction and in some ways, too speculative too. I think it would be best to condense some of this text and comment more on if the 20-10 bout (having the lowest interval duration and time > 80 %HRmax) is feasible and indicative of HIIE exercise vs. the other 3 regimens used. Also I believe that some of this text needs to be substituted by data from similarly habitually active participants rather than mice or trained cyclists, who have different exercise tolerance, BLa accumulation, etc. https://hes32-ctp.trendmicro.com:443/wis/clicktime/v1/query?url=https%3a%2f%2fpubmed.ncbi.nlm.nih.gov%2f28737586%2f&umid=2ba797c4-0763-42b4-9ef0-d2ada0e77048&auth=768f192bba830b801fed4f40fb360f4d1374fa7c-583789331f56d08dc8e7493023e56163af122adf

We feel this section is more impactful and clearer has part of the development of the discussion. But agree it was repetitive of the text used in the introduction, therefore we have edited the introduction to summarise the literature to prevent duplication (Line 62-70). 

I also believe you need to talk about the fact that the 10 X 1 regimen is at a fixed intensity; whereas, the other protocols are all-out or self-paced. Thus, the first regimen is imposed upon each participant; whereas, in the other 3 sessions, the exerciser has total control of his/her effort exerted. There is work showing that this feature can alter perceptions of exercise, so perhaps a few lines of text needs to be included here acknowledging this attribute.

We agree that this is an important distinction between the 4 protocols used within the current study. As such, we have added a short discussion of this point to lines 384-391 of the discussion. In short, our aim was not to investigate if the intensity regime influenced perceptual responses, and the design employed does not allow this, as different exercise modalities where used. However, previous research has shown that self-selected intensities results in more negative affect (using the FS) and, therefore, the intensity regime could have influenced the current data. 

Reviewer #3: This is an interesting study examining acute physiological, perceptual and motivational responses to popular social media HIIT protocols in comparison to evidence-based HIIT protocols. I commend the researchers for their novel study, which is particularly timely given many people’s time at home has been significant during the past year and interest in social media based workouts has also increased.

The manuscript is very well written, with a strong and balanced discussion including key studies in this field and highlighting opportunities for future research.

We appreciate the time the reviewer has dedicated to improve our manuscript and thank them for the feedback provided. 

You may wish to consider the points below:

Methods:

It would be useful to include further details regarding the four HIIT protocols. For example, where were the social media HIIT sessions completed? In the lab? Details of a warm-up were provided, however did participants also complete a cool-down?

More detail about the conditions in which the HIIT protocols were carried out have been added to the Methods section (Line 110-114). 

In addition to the popularity of the YouTube clips, what considerations were made when choosing these two HIIT workouts?

We thank the reviewer for this suggestion. A new paragraph within the methods under the heading of ‘Training Protocols’ now provides greater insight into why the protocols were selected. In brief, we used 4 criteria to assess videos found on YouTube 1) had to be featured on a popular YouTube fitness channel 2) have ‘HIIT’ in the title of the video 3) take less than 20 minutes, to take advantage of the time-saving nature of HIIT 4) include body weight exercises with no equipment. From these videos the SM-20:10 protocol was included as the video uses “Tabata training”, a variation of the original protocol designed by Tabata et al. (1996) which has been demonstrated to increases in VO2peak. SM-40:20 was included as the protocol used a blend of aerobic and resistance-based exercises (e.g. press-ups) which would not typically fall under the traditional definition of HIIT, but is used by a number of videos found on social media channels.

Discussion:

It might be useful to consider the venue in which HIIT sessions were conducted when explaining findings. Enjoyment and motivation may differ for a lab based session in comparison to other venues (e.g. home, gym, outdoors, etc.). In addition, the variety of exercises included for the social media and BW HIIT protocols, in comparison to using only the cycle ergometer, may also explain differences in enjoyment and motivation. The age of participants might also be considered, as younger adults may find social media based PA approaches more acceptable and relevant than other age groups.

We agree that the environment may have impacted the perceptual responses to the exercise protocols, this has now been added to the limitation sections. Equally the age of the participants is relevant to the responses seen as we agree social media protocols are aimed at a younger age group as a way to increase participation. This has been added to the limitation section (Line 433). 

Limitations:

Participants being classified as recreationally active has been noted as a limitation of the study, however the sample size has not been mentioned.

The sample size has been added to the limitations, however to our knowledge this is one of the largest cross-over acute HIIT studies conducted.

---

## [Decision Letter · Decision Letter 1]

8 Sep 2021

Evidence-based vs. social media based high-intensity interval training protocols: physiological and perceptual responses

PONE-D-21-08687R1

Dear Dr. Cocks,

We’re pleased to inform you that your manuscript has been judged scientifically suitable for publication and will be formally accepted for publication once it meets all outstanding technical requirements.

Kind regards,

Matthew M. Schubert, Ph.D.

Academic Editor

PLOS ONE

Additional Editor Comments (optional):

Reviewers' comments:

Reviewer's Responses to Questions

**Comments to the Author**

1. If the authors have adequately addressed your comments raised in a previous round of review and you feel that this manuscript is now acceptable for publication, you may indicate that here to bypass the “Comments to the Author” section, enter your conflict of interest statement in the “Confidential to Editor” section, and submit your "Accept" recommendation.

Reviewer #2: All comments have been addressed

Reviewer #3: All comments have been addressed

2. Is the manuscript technically sound, and do the data support the conclusions?

Reviewer #2: Yes

Reviewer #3: Yes

3. Has the statistical analysis been performed appropriately and rigorously? 

Reviewer #2: Yes

Reviewer #3: Yes

4. Have the authors made all data underlying the findings in their manuscript fully available?

Reviewer #2: Yes

Reviewer #3: Yes

5. Is the manuscript presented in an intelligible fashion and written in standard English?

Reviewer #2: Yes

Reviewer #3: Yes

6. Review Comments to the Author

Reviewer #2: I appreciate the revisions made by the Authors as well as the thorough rebuttal developed to my initial concerns--good luck with this line of work!

Reviewer #3: Thank you for addressing all comments thoroughly in your response letter; I have no further queries.

7. PLOS authors have the option to publish the peer review history of their article (what does this mean?). If published, this will include your full peer review and any attached files.

Reviewer #2: No

Reviewer #3: No

---

## [Editor Report · Acceptance letter]

21 Sep 2021

PONE-D-21-08687R1 

Evidence-based vs. social media based high-intensity interval training protocols: physiological and perceptual responses 

Dear Dr. Cocks:

I'm pleased to inform you that your manuscript has been deemed suitable for publication in PLOS ONE. Congratulations! Your manuscript is now with our production department. 

Kind regards, 

on behalf of

Dr. Matthew M. Schubert 

Academic Editor

PLOS ONE